# Neutrophil Activated by the Famous and Potent PMA (Phorbol Myristate Acetate)

**DOI:** 10.3390/cells11182889

**Published:** 2022-09-16

**Authors:** Hylane Luiz Damascena, Wendy Ann Assis Silveira, Mariana S. Castro, Wagner Fontes

**Affiliations:** Laboratory of Protein Chemistry and Biochemistry, Department of Cell Biology, Institute of Biological Sciences, University of Brasília, Federal District, Brasilia 70910-900, Brazil

**Keywords:** neutrophil, PMA, NETs, PAD4, neutrophil extracellular trap, citrullination, PKC

## Abstract

This review will briefly outline the major signaling pathways in PMA-activated neutrophils. PMA is widely used to understand neutrophil pathways and formation of NETs. PMA activates PKC; however, we highlight some isoforms that contribute to specific functions. PKC α, β and δ contribute to ROS production while PKC βII and PKC ζ are involved in cytoskeleton remodeling. Actin polymerization is important for the chemotaxis of neutrophils and its remodeling is connected to ROS balance. We suggest that, although ROS and production of NETs are usually observed together in PMA-activated neutrophils, there might be a regulatory mechanism balancing both. Interestingly, we suggest that serine proteases might determine the PAD4 action. PAD4 could be responsible for the activation of the NF-κB pathway that leads to IL-1β release, triggering the cleavage of gasdermin D by serine proteases such as elastase, leading to pore formation contributing to release of NETs. On the other hand, when serine proteases are inhibited, NETs are formed by citrullination through the PAD4 pathway. This review puts together results from the last 31 years of research on the effects of PMA on the neutrophil and proposes new insights on their interpretation.

## 1. Introduction

Neutrophils are the first cells to migrate to an inflammation site. They fight against pathogens using phagocytosis, reactive oxygen species (ROS), enzymes and DNA release in neutrophil extracellular traps (NETs) [1,2,3].

It is through pattern recognition receptors (PRRs) that neutrophils recognize molecular structures such as pathogen-associated molecular patterns (PAMPs) or damage-associated molecular patterns (DAMPs) and start the activation process [3]. Neutrophil activation and recruitment are regulated by pro-inflammatory cytokines, endothelial cells, platelet activation factor (PAF), and microbial products, among other substances [4,5,6], but that is not the case for PMA.

Phorbol 12-myristate 13-acetate (PMA) is a phorbol compound obtained from *Croton tiglium*, a plant of the Euphorbiaceae family. It was first observed to be a tumor promoter in mouse skin, then other studies showed that the carcinogenic effect was correlated to the proinflammatory activity and to protein kinase C (PKC) activation, producing a diversity of biological effects. Nowadays, PMA is widely used in experiments to promote tumor development and leukocyte activation [7,8]. Although it is an artificial stimulus, it is widely known and commonly used to study the neutrophil, because PMA remarkably induces pathways similar to other stimuli, with the participation of PKC, myeloperoxidase (MPO) and neutrophil elastase (NE) [7,8]. In addition, PMA is known to induce neutrophil degranulation, metabolic changes, ROS production and NET formation, in nanomolar concentrations [9,10,11,12]. PMA is widely used to mimic PKC regulation by the same mechanism as diacylglycerol (DAG), a lipidic second messenger generated by the enzyme phospholipase C (PLC) when catalyzing the degradation of phosphatidylinositol 4, 5 bisphosphate (PIP2). DAG is responsible for the activation of PKC [9].

It is known that PMA activates PKC, however neutrophils express different PKC isoforms, contributing to different functions. Here, we highlight that while α, β and δ isoforms are involved in the translocation and phosphorylation of the NADPH oxidase components [10], and contribute to ROS production [11,12,13], PKC βII and PKC ζ participate in adhesion and migration [13,14,15]. In addition, ROS acts on neutrophil migration at the same time, impairing it through the oxidation of actin [16,17]. PKC δ activates the phosphoinositide-3-kinases (PI3K) pathway through myristoylated alanine-rich C-kinase substrate (MARCKS) phosphorylation. PtdIns (3,4,5) P3 (PIP3), a product of the PI3K pathway, activates Akt that contributes to p47^phox^ phosphorylation, however its contribution to the major death mechanism activated by PMA, a process named NETosis, is unclear. At the same time, PIP3 can activate PKC ζ, which activates protein arginine deiminase 4 (PAD4) promoting the release of NETs. However, there is NETs formation without histone citrullination. This review highlights relations among these aspects and brings together for the first time the two existing ways that PAD4 can contribute to the formation of NETs.

Neutrophil pathways activated by PMA have been well studied, but there are aspects that are not clear. As we did not find a previous comprehensive review about the effects of PMA on neutrophils, and due to the increased number of publications since the 1990s (Figure 1), we also put together an overview of the functional features and their respective molecular pathways that were rather scattered in the literature over the past 31 years, not excluding the most relevant publications before this period. This review will provide a focus on the available knowledge of PMA-activated neutrophil responses, as well as highlight the open questions and disagreement in the literature.

## 2. Neutrophils Are Activated by PMA

PMA crosses the plasma membrane without the aid of a membrane receptor and directly activates PKC by translocation of the cytosolic isoforms to the cell membrane [18,19].

### 2.1. Protein Kinase C

Protein kinase C (PKC) is a member of the serine/threonine kinases protein family. This family includes protein kinase A (PKA) and protein kinase B (PKB). The PKC isoforms are classified according to their regulatory domain that is variable among the three PKC subgroups. The conventional PKCs (cPKCs) are made up of four isoforms (α, βI, βII, γ) that are activated by DAG/PMA and are calcium-ion dependent. The novel PKCs (nPKCs) are made up of four isoforms (δ, θ, ξ, η), also activated by DAG/PMA, however they are calcium-independent. The atypical PKCs (aPKCs) are made up of two isoforms (ζ, ι/λ) which are DAG/PMA and calcium independent. They can also be activated by phosphatidylinositol (3,4,5)-trisphosphate (PIP3) or ceramide. In addition, all three groups need the phospholipid, phosphatidylserine, for activation [9].

Neutrophils express five PKC isoforms (α, βI, βII, δ, ζ) which have been suggested to participate in and regulate different functions such as adhesion, migration, degranulation, phagocytosis, apoptosis and ROS production by activation of the NADPH oxidase enzyme complex [20,21].

### 2.2. Protein Kinase C Isoform Locations and Translocation

Protein kinase C isoforms in unstimulated neutrophils are located in the cytosol and translocate to the plasma membrane upon activation by PMA [22], except PKC β isoforms which are partially located in the membrane [21]. Unlike nPKC δ, which has already fully translocated to the plasma membrane within two minutes [21], cPKC α [21,23], βI and βII are still initiating translocation upon activation [15,21,23]. Although the atypical PKC (aPKC) ζ has been cited for being independent of PMA, it exhibited a rapid translocation, within five minutes, to membrane fragments in PMA-activated cells [15]. PKC ζ was first detected in 1994 by Dang et al. and its activation by PMA is not direct, but happens downstream via PIP3 [9] (Figure 2). It is also implicated to have a role in neutrophil effector functions, such as adhesion [15,24]. However, there is a disagreement about PKC ζ acting as an effector of ROS production. Laudanna et al. (1998) and Dang et al. (2001), using an inhibitor based on myristoylated peptides from the pseudosubstrate region of PKCζ did not show effect on ROS production [13,25], while Kato et al. (2005) showed decrease of ROS consequent to the inhibition of PKC ζ [26]. 

The translocation of PKC δ leads to the phosphorylation of myristoylated alanine rich protein kinase C substrate (MARCKS) [21,27]. MARCKS translocates from the cytosol to the plasma membrane, which alters the mobility or affinity of the CD18 (β2 integrin), polymerizing actin, and therefore leading to cell degranulation, adhesion and migration [28].

### 2.3. PMA Triggers Degranulation

Cytoplasm of neutrophils contains granules and vesicles that have hundreds of proteins. The primary or azurophilic granules have about 69% of the total neutrophil myeloperoxidase, besides being rich in α-defensin, myeloblastin, elastase, azurocidin and cathepsin G [29]. The secondary or specific granules have about 54% of the gelatin-associated lipocalin (NGAL), 52% of the lactoferrin, and contain other proteins such as pentraxin 3 and haptoglobulins [29,30]. Both release antimicrobial proteins, while the tertiary or gelatinase granules are rich in gelatinase, a matrix metalloproteinase that degrades the extracellular matrix. In addition, after fusion with the cell membrane during degranulation, granular membrane proteins expose receptors for adhesion and migration [29,31,32]. The secretory vesicles, which contain proteins such as the Toll-like receptor (TLR) 2, 4 and 8, as well as β and α-actin, add proteins to the cell membrane to recognize PAMPs and for migration [29,33].

Initially, neutrophils activated by PMA release the content of secretory vesicles, tertiary, secondary and primary granules already in the first minutes, lasting for hours [34,35]. *PKC δ* gene deficient neutrophils showed markedly reduced lactoferrin degranulation following stimulation with PMA [36] and neutrophils preincubated with a MARCKS synthetic peptide showed increased degranulation [37]. The facts that adhesion receptors get exposed after degranulation and that MARCKS is involved in both processes reinforces the correlation between degranulation and adhesion, even though degranulation happens independent of an adhesion surface [38]. Downstream to the same pathway, Karlsson et al., 2000 showed that PI3K inhibition does not change the degranulation of any type of granules or secretory vesicles [34]. On the other hand, Chen et al., 2010, showed that *Akt2* gene deletion decreased the primary granule degranulation, suggesting a role for Akt in degranulation, independent of PI3K [39]. Furthermore, PMA was ineffective at eliciting primary granule content release in Rac2 knockout (KO) mice, however it still induced the release of secondary granules [40]. Interestingly, the suppression of both pathways, Akt2 and Rac2, affect the release of primary granules and extracellular ROS formation [39,40], while PI3K inhibition does not change the degranulation of any type of granules; it does however, inhibit intracellular ROS [34]. After NADPH oxidase assembly, the specific granules have the necessary components for ROS production, with the exception of myeloperoxidase to convert hydrogen peroxide (H_2_O_2_) to hypochlorous acid (HOCl) [41,42,43]. As degranulation of primary granules, as well as extracellular ROS production are decreased in Akt2 or Rac2 deletion [39,40], this suggests that the NADPH oxidase at the cell membrane and at secondary granules might be impaired, and that the primary granules might contribute to the amount of extracellular ROS. In 2016, Björnsdottir et al. proposed a fusion of specific granules and azurophil granules, producing intracellular ROS in the absence of phagosome formation [44]; however, it seems that specific granules do not contribute as much to intracellular ROS as opposed to extracellular ROS, perhaps due to the incorporation of their NADPH oxidase to the cell membrane after degranulation [45,46]. Based on these publications, we suggest that there are two different pathways to intracellular ROS influenced by PI3K, while extracellular ROS is Akt- and Rac2-dependent with the possible participation of the primary granules.

### 2.4. Crosstalk between Adhesion/Migration and NADPH Oxidase

PMA activates an increase of the main adhesion molecules of neutrophils: LFA-1 (CD11a/CD18) [47], MAC-1 (CD11b/CD18) [48,49], ICAM-1 (CD54), complement receptor 1 (CR1,CD35) at the plasma membrane [50], and promotes L-selectin shedding, increasing soluble L-selectin [49,51]. Moreover, PMA triggers an increase of E-selectin on endothelial cells [52], while decreasing P-selectin glycoprotein ligand-1 (PSGL-1) on the surface of neutrophils, reducing their adhesion to P-selectin [53] on the surface of endothelial cells. These mechanisms of regulation are involved in the control of neutrophils’ rolling velocity on the endothelium, favoring their diapedesis. Therefore, PMA promotes actin polymerization [54], supporting the rearrangement of the cytoskeleton, which is important for the assembly of the NADPH oxidase complex [55]. Accordingly, ROS production also allows neutrophil adhesion. Shen et al. (1999) showed that a decrease in ROS reduces approximately 60% of neutrophil adhesion [56]. Another antiadhesive molecule, CD43 (leukosialin), binds to E-selectin, limiting neutrophil interaction with the endothelium [57] and is decreased on the surface of PMA-activated neutrophils [58]. The serine protease inhibitor (3,4-DCI) prevents the increase in ROS production, as well as it inhibits the decrease in CD43 activated by PMA [59]. The decrease in CD43 occurs because it undergoes juxtamembranous cleavage by a serine protease, cathepsin G [60], and is relevant to cell adhesion, which is limited by CD43.

The rearrangement of the cytoskeleton enables the activation of structural and regulatory proteins, such as actin, moesin and cofilin, that interact with the cytosolic components p47^phox^, p40^phox^ and p67^phox^ of the NADPH oxidase enzyme complex. This interaction is thought to enable the translocation of these cytosolic components to the plasma membrane, therefore allowing the binding of these components to the membrane bound subunits, gp91^phox^ and gp22^phox^, which together form the flavocytochrome b558 subunit of this complex [61,62]. Interestingly, cytoskeletal components have been indicated in the regulation and organization of the NADPH oxidase complex, as well as the complex itself having a regulatory role in cytoskeletal conformation [63]. Actin polymerization contributes to the activation of the NADPH oxidase complex [64], allowing translocation of the components [62,65]. Nevertheless, NADPH oxidase-generated ROS reduces actin polymerization through actin glutathionylation, impairing neutrophil chemotaxis [17]. On the other hand, total inhibition of ROS leads to a more frequent formation of multiple pseudopodia and a loss of direction during migration [66]. This suggests that ROS contributes to the adhesion and migration of neutrophils by orchestrating the shedding of adhesion molecules and actin polymerization, although the ideal concentration of ROS remains unclear.

Actin polymerization is the consequence of a remarkable association between PKC isoforms [67] and the cytoskeleton. Although Downey et al. (1992) showed that actin assembly was not inhibited by conventional PKC inhibitors [68], Niggli et al. (1999) showed that the inhibitor of actin polymerization (latrunculin A) reduced cytoskeletal PKC βII by 43%, suggesting a physical binding between PKC βII and F-actin [14]. In addition, the fact that PKC ζ inhibited PMA-activated adhesion and not ROS production suggests its possible effector function to trigger pathways leading to adhesion and not ROS [13,15,26], reinforcing the idea that the regulation of cytoskeleton remodeling is established by these isoforms of PKC.

### 2.5. NADPH Oxidase Subunit Phosphorylation and Translocation

Apart from the suggested role of the cytoskeleton in the translocation of NADPH oxidase cytosolic subunits, other events are necessary to trigger the full activation of this enzyme complex, such as phosphorylation of the oxidase subunits, as well as the activation of the Rho GTPase Rac2, a non-specific participant in this process [19]. Phosphorylation of the NADPH oxidase components by PKC isoforms is essential for ROS production in the PMA-activated neutrophil [10,69,70,71,72,73]. PKC isoforms α, β and δ have been shown to either activate or inhibit p47^phox^ depending on the phosphorylation site [74,75], and hyperphosphorylation of p47^phox^ leads to the deactivation of the NADPH oxidase complex [76]. Previously, in 2002, Fontayne, et al. described how PKC isoforms α, βII, δ and ζ played roles in the translocation of p47^phox^ and its subsequent binding to the gp22^phox^ membrane subunit in a cell free system [10]. Additionally, they demonstrated that each PKC isoform could individually phosphorylate p47^phox^, therefore inducing activation of the NADPH oxidase complex. It was noted, however, that the ζ isoform phosphorylated fewer and different serine sites in comparison with the other isoforms (α, βII and δ), which resulted in a higher level of NADPH oxidase activity. They further suggested that the PKC isoforms α, βII and δ could be associated with attenuated oxidase activity, therefore indicating specific roles for these PKC isoforms [10]. In summary, different PKC isoforms are related to specific phosphorylation patterns of the p47^phox^, leading to the modulation of ROS production in the first minutes after PMA activation [11,77,78].

Translocation patterns of PKC isoforms α and βII coincide with the translocation pattern of p47^phox^, adding evidence to their participation in NADPH oxidase assembly. This was not the case with PKC δ. Even though it has a different translocation profile to p47^phox^, PKC δ translocates faster than PKC α and βII [23]. However, specific inhibition of PKC δ decreases 96% of ROS production, and the immunodepletion of PKC δ decreases approximately 50% of intracellular ROS [12]. Recently, PKC δ tyrosine 155 phosphorylation was demonstrated not to be required for ROS production in PMA-activated neutrophils [79]; however, PKC δ also presents serine/threonine phosphorylation sites [9]. Moreover, PKC δ phosphorylates NAD kinase (NADK) which is required for the synthesis of NADPH, the substrate of the oxidase complex [80]. Another study in 2000, by Dekker et al. demonstrated that PKC β is responsible for at least 50% of superoxide production. Interestingly, it was demonstrated that an increased concentration (1 µM) of the inhibitor 379196, a specific inhibitor of PKC β, decreased ROS production [11]. Besides that, PKC α inhibition abrogated ROS [13], which seemingly confirms the role of these isoforms in NADPH oxidase activation and assembly; however, it is not clear which PKC isoform is most important to ROS production. Moreover, as the sum of percentages assigned to each isoform clearly surpass 100%, it is likely that there is a crosstalk among the PKC isoform mechanisms.

Apart from the phosphorylation of the NADPH oxidase complex, PKC isoforms have been shown to phosphorylate the Raf/MEK/ERK/p38/Rac-2 pathways [81,82,83,84] that also phosphorylate NADPH oxidase complex components (Figure 2). The MAPK pathway is composed of three kinds of kinases, MAPK kinase kinase (e.g., Raf), MAPK kinase (e.g., MEK), and MAPK (e.g., ERK1/2, p38, JNK), that are sequentially activated by phosphorylation [85]. Three MAPK families have been defined in mammalian cells, classical MAPK or extracellular signal-regulated kinases 1/2 (ERK1/2), C-Jun N-terminal kinase/stress activated protein kinase (JNK/SAPK) and p38 kinase [86]. The reduction in abundance of ERK/MAPK leads to NADPH oxidase inhibition [87,88,89] because MEK 1/2 phosphorylates the p67^phox^ subunit of NADPH oxidase complex [90], and p38 MAPK phosphorylates the p47^phox^ component [91]. Moreover, ERK2 [92] and PKC δ phosphorylate MARCKS [21], which translocates to the plasma membrane after phosphorylation [93] and activates PI3K pathway through the release of PIP2 molecules that recruit multiple active PI3K to the membrane surface [94]. Active PI3K at the membrane surface, exposed to PIP2, will phosphorylate it to PIP3. Furthermore, PI3K inhibition did not affect the MAPK pathway activated by PMA [95]. Therefore, we propose that MAPK pathway occurs concomitantly to the PI3K pathway.

## 3. PI3K Participates in ROS Production

Phosphoinositide-3-kinases (PI3K) are a family of lipid kinases that are grouped into classes I, II and III. Class I PI3K is the most widely studied and is divided into IA (PI3Kα, PI3Kβ, PI3Kδ) and IB (PI3Kγ) [96,97]. PKC β isozymes phosphorylate the PI3Kγ isotype [98], which phosphorylates PtdIns (4,5) P2 (or PIP2) to PtdIns (3,4,5) P3 (or PIP3). This leads to the translocation of various proteins through the pleckstrin homology (PH) domain [99]. Proteins such as serine/threonine kinases Akt/PKB, which are involved in the rearrangement of the cytoskeleton in stimulated cells, contain the PH domain that binds to PIP3. This binding leads to Akt activation [100]. The phosphorylation of ERK by PI3Kγ also leads to ROS generation, apoptosis inhibition, and chromatin decondensation [101]. Thus, isoform PKC β triggers PI3Kγ pathway, which also activates ERK, contributing to functions like ROS production (Figure 2).

PI3K participates in the intracellular ROS production by the NADPH oxidase complex [34,42,102,103]. Class III PI3K phosphorylates PtdIns to PtdIns3P (PI3P), described to regulate lysosomal trafficking and to control autophagy. Interestingly, PI3P contributes to the first phase of intracellular ROS production by binding to the subunit p40^phox^ of the NADPH oxidase [104,105]. Class II PI3Ks also phosphorylate PtdIns to PtdIns3P, however their function has been little explored [96]. Class I and II PI3K have an intrinsic relation to ROS production, which includes the later activation of proteins such as Akt, although the Akt2 isoform is PI3K-independent [39]. Akt also phosphorylates mTORC, inhibits anti-apoptotic proteins and activates the NF-κB complex [100]. However, more studies are necessary to elucidate the functions of Class II PI3K.

Furthermore, class I PI3Ks activity is modulated by families of phosphoinositide phosphatases that include the tumor suppressor protein phosphatase and tensin homolog deleted from chromosome 10 (PTEN), SH2-containing inositol polyphosphate 5-phosphatase 1 (SHIP) and inositol-3,4-bisphosphate 4-phosphatase (INPP4A/4B) [99]. PTEN is a phospholipid phosphatase that dephosphorylates PIP3 and converts it to PIP2, antagonizing the PI3K activity. Thus, PTEN modulates various cell functions through dephosphorylation including the regulation of cell growth, adhesion, chemotaxis, and migration [106]. Besides that, PTEN participates in ROS production [107], because PIP2 is cleaved by phospholipase C (PLC) into DAG and inositol-triphosphate (IP3) that participate in PKC activation and the release of intracellular calcium, respectively [19] (Appendix A). Interestingly, Teimourian and Moghanloo (2015) showed that PTEN also contributes to NETs production [108]. They showed that PTEN down-regulation in neutrophils differentiated from HL-60 cells, activated by PMA, reduced 26% of NETs production. In addition, it was observed that PTEN depletion leads to an increase in PIP3 levels and improves ROS production [99]. Such depletion impairs PIP2 formation, therefore impacting the production of DAG and IP3 by phospholipase C (PLC) [109]. As a consequence, the calcium release induced by IP3 and PKC activation induced by DAG would also be downregulated, likely decreasing NETs formation. On the other hand, the reason for the increased PIP3 levels not compensating for NET production remains an open question.

### The Relation between Calcium and PMA Activation

Calcium is an intracellular signaling ion important for neutrophil adhesion and ROS production [110]. However, there is disagreement in the literature about PMA activated calcium influx and efflux levels and its contribution to ROS and NETs formation. Shen et al. (1999), using fluorescence spectrophotometry, showed that PMA increases intracellular calcium by less than 20% in the first seconds after stimulation [111]. Although PMA activates endoplasmic reticulum stress [112], some authors do not consider the influx and efflux of calcium by PMA to be significant, rather only a minimal oscillation in calcium levels early after stimulation [80,112,113,114]. The occurrence of this oscillation in calcium levels together with ROS production suggests its involvement in NETosis [83]. It has been shown that the depletion of calcium stores by thapsigargin affects ROS production [115,116], while reduction of ROS by chelating extracellular calcium with EGTA is not observed [115,117].

Furthermore, calcium ion is associated with phospholipases A2 (PLA_2_) and D (PLD) activity, possibly involved in ROS production [118,119]. PLA_2_ is phosphorylated by PKC and ERK1. When activated, it cleaves membrane phospholipids to produce arachidonic acid, which is released by neutrophils activated by PMA [120,121]. Meanwhile, phospholipase D, which hydrolyzes phospholipids to phosphatidic acid (PA) and choline [122], is phosphorylated by nPKC, leading to its activation in response to PMA [123]. PLD is regulated by ADP-ribosylation factor (ARF-1) [124], which also regulates actin assembly allowing vesicle fusion [125].

The decrease of intracellular calcium impairs PLA_2_ and PLD [122,126], while extracellular calcium has no effect on PLA_2_ [126]. However, there is a study suggesting that PLA_2_ is not involved in NADPH oxidase activity, but influences ROS by affecting granule fusion [119].

On the other hand, Gupta et al. (2014) and Suh et al. (2001) showed that chelation of extracellular calcium reduces ROS by approximately 60% [127,128]. Moreover, Gupta et al. (2014) showed that for efficient NETosis to occur there must be a mobilization of intra-and extracellular calcium pools [127]. Accordingly, Parker et al. (2012) and Kenny et al. (2017) showed that the intracellular calcium chelator, BAPTA-AM, reduced ROS-dependent NETs in PMA stimulation [2,129]. Conflictingly, Douda et al. (2015) suggest that the absence of extracellular calcium had no effect on NETs release based on cells cultured without calcium [130]. Although calcium could contribute to ROS production by PLA_2_ and PLD, the literature suggests that intracellular calcium is not decisive to ROS production activated by PMA, because it does not induce a significant intracellular calcium mobilization. Meanwhile, the participation of extracellular calcium and the contribution mechanisms of both stores to NETs formation remain unclear, and it is still necessary to identify whether PMA activation is expressively affected by the extracellular calcium concentration.

## 4. Neutrophil Extracellular Trap (NET) Formation

NETs are a mechanism used by neutrophils, in which they release DNA and proteins to promote the extracellular capture and death of microorganisms. However, they are considered double-edged swords for being associated with the cause and severity of some diseases, such as thrombosis, autoimmune diseases, and cancers. Recently, it was also associated with the severe acute respiratory syndrome observed in COVID-19 [131,132]. A better understanding of the pathways related to NETs is needed so that more effective therapeutic approaches to these diseases can be developed [133].

NETs is a process that occurs independently of phagocytosis [1,134,135]. Currently, NETs are classified as suicidal or vital. Suicidal NETs are ROS-dependent and vital NETs can be triggered by ROS-dependent or ROS independent mechanisms [133,136]. PMA is able to induce ROS-dependent suicidal NETs [1,84]. Suicidal NETosis is a death process that requires 3–4 h for the complete formation of NETs, although some cells already show formation within 30–60 min after activation [84,127,137,138]. In this mechanism, NADPH oxidase is activated via PKC/Raf/MERK/ERK, that also leads to the activation of the enzyme peptidylarginine deaminase 4 (PAD4), which modifies histones, converting arginine (Arg) into citrulline (Cit). Such modification promotes the loss of positive charges, preventing their interaction with DNA, leading to chromatin decondensation [139,140,141].

Hydrogen peroxide is also responsible for activating and dissociating a complex of proteins present in azurophilic granules (inhibition of such proteases by SerpinB1 limits NETs formation) [142] Then, H_2_O_2_, together with MPO, activates NE, which degrades F-actin allowing the translocation of NE to the nucleus [143], promoting a partial degradation of histones [144]. In addition, F-actin glutathionylation triggered by ROS also leads to actin degradation [145], promoting structural changes that, together with the mechanisms described above, culminate in chromatin swelling and decondensation [2,83,146] (Figure 2). This allows NETs to be released via nuclear budding or vesicle formation in vital NETs [81] or via rupture of the nuclear membrane and finally of the plasma membrane in suicidal NETs. Recently, the nuclear component lamin B was described to be disassembled after phosphorylation by PKC α resulting in nuclear rupture [147].

Even though PI3K inhibition does not reduce NETs formation [148], this pathway participates in NETs formation by means of its contribution to ROS production. There is still disagreement about whether the Akt pathway contributes to NETs formation. Starting from evidence that PMA induces Akt phosphorylation [84], Douda et al. (2014) suggested that ROS-dependent NETosis in PMA (25 nM) stimulated neutrophils requires Akt activation, using both Akt and NOX2 inhibitor assays [149]. On the other hand, DeSouza-Vieira et al. (2016) rejected that hypothesis, as they found that the Akt inhibitor (124005) had no effect on NETs release when cells were activated with 100 nM PMA [101]. The different methods of these studies do not allow us to do an appropriate comparison, thus, more studies are needed to elucidate this.

### 4.1. Protein Arginine Deiminase 4 (PAD4)

The activation of PAD4 induced by PMA leads to chromatin decondensation [84,146,150]. The chromatin structure is organized with two of each histones H3, H4, H2B and H2At in the DNA to form a nucleosome core particle, and a linker formed by DNA and H1 [151]. Peptidylarginine deiminases (PADs) are a family of enzymes that are located in the nucleus and modify histones by converting arginine (Arg) to citrulline (Cit) [140,141]. This decreases the positive charges along the histone termini, reducing electrostatic interactions with DNA, and thus allowing chromatin to assume an extended conformation [141].

There are five highly conserved PADs in mammalians, namely PAD1, 2, 3, 4 and 6 [151]. Neutrophils express high levels of PAD4 [152] which mediates chromatin decondensation and NETs formation [153], ejecting 15–25-nm chromatin fibers [1]. Li et al. (2010) showed that mouse neutrophils activated by PMA have 48.5% of histone citrullination [154], while Holmes et al. (2019) identified approximately 20% in humans [155]. Other evidence in the literature shows further divergence whether NETs induced by PMA are citrullination-dependent or independent. Tatsiy e McDonald (2018) showed that GSK484, a specific inhibitor of PAD4, abrogated the citrullination of histone H3, impairing NETs formation [84] and Hawez et al. (2019) detected the PMA-induced increase in PAD4 and in citrullination of histone H3 by western blotting [156]. Oppositely, Neeli and Radic (2013) and Konig and Andrade (2016) did not detect citrullination of histone H3 by western blotting after 2 h of PMA stimulus [140,150]. Neeli and Radic (2013) demonstrated the inhibition of PAD4 by PKC α and its activation by PKC ζ isoforms, but the dominant isoform after PMA stimulation is the alpha, coherent with the lack of citrullination in the NETs [150]. In that regard, Kenny et al. (2017) did not observe impairment on NETs by inhibition of PAD2 (Cl-amidine) and PAD4 (BB-Cl-amidine and TDFA) [129]. In 2018, de Bont et al. showed that citrullination activated by PMA was detected in NETs after inhibition of serine proteases [83]. They suggested that serine proteases promote the cleavage of citrullinated histones, possibly explaining the previous disagreement. Based on these results, and on the results discussed below about PAD4 citrullination of NF-κB p65 (see the topic NF-κB), we suggest that PAD4 may influence NET formation by at least two pathways, based on histone H3 citrullination or IL-1β release and gasdermin cleavage triggered by activity of serine proteases. This second pathway possibly explains the presence of NETs in the absence of citrullinated histones, as the same proteases might cleave the citrullinated histones (see Gasdermin D topic) (Appendix A).

### 4.2. Autophagy

Autophagy is a catabolic process that plays important roles in preventing damage to cells, promoting cell survival during nutrient deprivation and oxidative stress [157,158,159]. During phagocytosis, ROS produced by NADPH oxidase or the activation of AMPK generated by ATP depletion leads to the activation of autophagy [160,161].

The induction of autophagy occurs via autophagy related (ATG) proteins. There is a cooperation between two protein complexes, one is composed by a serine/threonine protein kinase (ULK1) interacting with ATG101 and ATG13. The other complex contains PI3K and ATG14 [162]. PI3K catalyzes the production of PIP3, which binds LC3B [163]. LC3B is then translocated to the autophagosome membrane [164] in the maturation process. Further protein interactions, such as with Rubicon and Rab7 stimulate the fusion of the autophagosome with endosomes/lysosomes [161,165].

The canonical autophagy pathway is modulated by the activation of mTORC1 to phosphorylate ULK1 [160], while the induction of autophagy by phagocytosis is dependent on lysosome-associated membrane protein (LAP) [161].

Autophagy participates in some functions of the neutrophil, such as degranulation and ROS production [164]. Some studies showed that neutrophils activated by PMA release NETs with the involvement of autophagy as well as ROS [166], as the inhibition of either autophagy or ROS production prevented chromatin decondensation [167]. These studies used PI3K inhibition that leads to impairment of the NETs, but it also inhibits ROS production. Thus, the role of autophagy in NETs was not quite clear. However, Park et al. (2017) observed an increase of LC3I and LC3II induced by PMA, and the use of autophagy inhibitors eliminated what the authors named an autophagy priming for NETs [168]. Besides that, recently Germic et al. (2017) showed that Atg5 knockdown does not affect NETs production in neutrophils activated by GM-CSF, C5a, LPS and PMA, elucidating that autophagy is not mandatory for NETs production [169].

### 4.3. NF-κB

Nuclear factor-kappa B (NF-κB) is a family of transcription factors that consists of five proteins that share the amino-terminal Rel homology domain (RHD) [170]. The five proteins p65 (RelA), RelB, c-Rel, p105/p50 (NF-κB1), and p100/52 (NF-κB2) can associate among themselves, forming homo- or heterodimers in a more transcriptionally active structure. The p50/65 heterodimer is the most abundant form, being found in almost all cell types [171].

NF-κB can be activated in both the canonical and non-canonical pathways. In the canonical pathway, microbial products and proinflammatory cytokines trigger the activation of p65-containing complexes and it requires the IKKβ and IKKγ kinases. On the other hand, in the non-canonical pathway, which probably works together with the canonical, different cytokines trigger the activation of p52-containing complexes. Both pathways lead to the translocation of different NF-κB heterodimers to the nucleus [172,173].

In neutrophils, NF-κB promotes cell adhesion, inflammation and apoptosis. Contrary to the LPS stimulation, which inhibits NF-κB in neutrophils, leading to apoptosis [174], PMA stimulus activates the canonical pathway, translocating NF-κB p50/p65 to the nucleus [175]. p65 leads to NETs formation, while its inhibition decreases NETs [176]. Interestingly, during the inhibition of NF-κB, neutrophil survival could be sustained by p38 MAPK activation after autophosphorylation [177]. The p65 could be phosphorylated in serine (Ser), threonine (Thr), or tyrosine (Tyr) sites by kinases such as PKAc, MSK-1/MSK-2 and Pin1 [172]. In addition, p38 MAPK can regulate the NF-κB pathway [178]. Furthermore, PAD4 can citrullinate NF-κB p65 to promote its nuclear translocation and mediate IL-1β production (Figure 2) [179], which also induces NETs formation and enhances ROS production [180]. It is known that IL-1β can be released by caspase-dependent or independent pathways [181]. In the caspase-dependent pathway, there is activation via the canonical inflammasome pathway, while the non-canonical pathway, which is activated by PMA [182], is independent of the inflammasome. Both pathways allow for the cleavage of gasdermin D [183,184].

### 4.4. Gasdermin D

Recently, some studies showed the importance of yet another mechanism for NETs formation, the gasdermin D protein. Gasdermin D (GSDMD) has the N-terminal and the C-terminal domains separated by a binder portion. It is known to promote a type of cell death, called pyroptosis [183,184]. Gasdermin cleavage leaves the portion GSDMD-C in the cytosol, while the GSDMD-N product is oligomerized in the cell membrane forming pores [185]. In neutrophils activated by PMA, gasdermin was cleaved by ROS-activated serine proteases such as NE [182], allowing the release of IL-1β [181]. As mentioned above, IL-1β triggers NETs formation [180,186] and is involved in the process of recruiting other neutrophils to the infection site in response to large pathogens that cannot be phagocytosed [187], also entrapped in NETs [135]. PMA-activated neutrophils tend to be short-lived cells [188], due to apoptosis, pyroptosis [189] and NETosis, which is the main death mechanism [1,188,189,190,191], even though they utilize autophagy mechanisms [167]. However, Desai et al. (2016) showed that inhibition of necroptosis (RIPK1 and MLKL) and knockout of RIPK3 reduces NETs formation [192]. On the contrary, Amini et al. (2016) using knockout of RIPK3 [193], and Remijsen et al. (2011) who inhibited RIPK1 by necrostatin-1, did not observe effects on NETs formation [167]. Thus, gasdermin D-induced pore formation can be considered a common point among ROS production, NET formation, IL-1β release and NF-κB p65 pathways, although the mechanism for the latter is not quite clear in the literature yet. It could be activated by p38 MAPK and suffer modification by PAD4, to then participate in NETs formation by gasdermin D activation. However, it remains to be elucidated if this is actually a cascade, therefore connecting the upstream and downstream proteins, or if the pathways of gasdermin D cleavage and IL-1β release by NF-κB are distinct and concurrently performed.

## 5. Conclusions

Even though PMA is not an endogenous or microbial activator, it allows for the understanding of different pathways of neutrophil activation. PMA crosses the cell membrane independently of receptors and activates PKC α, β and δ. PKC δ phosphorylates MARCKS that leads to the activation of the PI3K pathway, while PKC β phosphorylates PI3K and MAPK pathways. Moreover, PI3Kγ activates ERK, and these pathways work together, all leading to the phosphorylation and translocation of components of the NADPH oxidase complex. The regulatory crosstalk between ROS production and cytoskeleton remodeling still requires further investigation, as these processes might be hubs for integrative responses of the neutrophil.

Much more detail is provided in the literature regarding the PKC, MAPK and PI3K pathways, raising possible interpretations, such as a regulatory mechanism balancing ROS and NETs production. However, some aspects are still open to elucidation, such as the role of calcium release and the Akt contribution to NETs production.

PKC ζ contributes to the citrullination of histones promoted by PAD4. However, we found disagreement in the literature regarding whether NETs are dependent or not on histone citrullination. Interestingly, PAD4 participates in other functions, such as NF-κB activation and the release of IL-1β. Moreover, IL-1β triggers the cleavage of gasdermin D by serine proteases such as elastase, leading to both pore formation and NETs formation. Although it is known that IL-1β recruits other neutrophils and is involved in the process of NETs formation, the pathways involved are not fully elucidated.

In all, even the effects of one of the most studied neutrophil activators still has a number of controversial or unclear aspects that we highlighted in this first present review.

## Figures and Tables

**Figure 1 cells-11-02889-f001:**
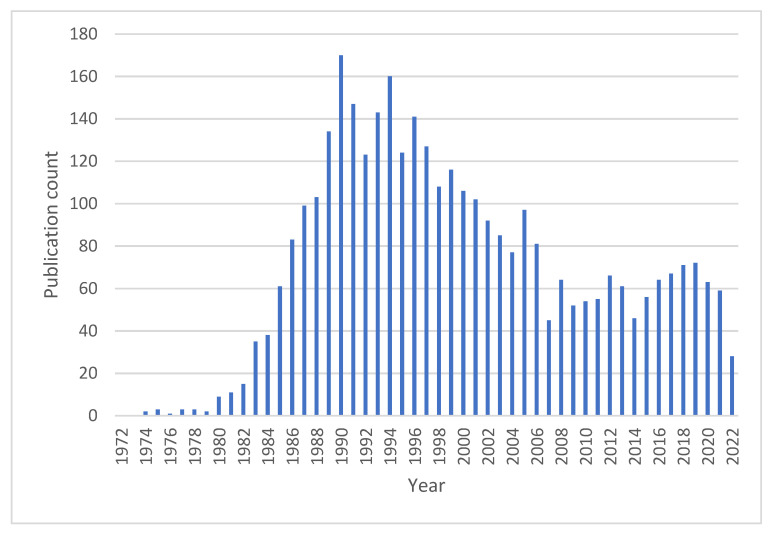
Distribution of articles containing the terms neutrophil and PMA along the last five decades, indexed to the PubMed library.

**Figure 2 cells-11-02889-f002:**
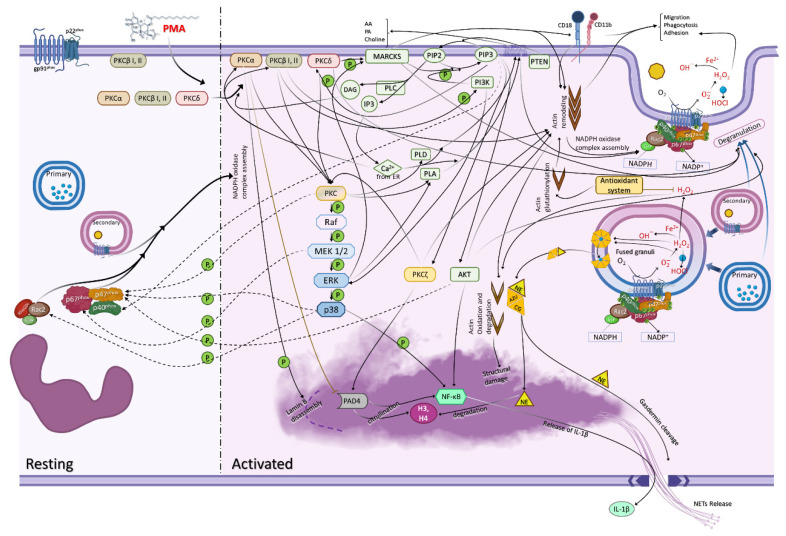
Major signaling pathways in PMA-activated neutrophils. PMA crosses the cell membrane and activates PKC isoforms (α, βI, βII, δ) which translocate to the cell membrane while some PKC β isoforms are already there. PKC δ phosphorylates MARCKS which activates the PI3K pathway allowing the phosphorylation of PIP2 to PIP3. PIP3, in turn, binds to PH domains such as the one found in Akt. PTEN regulates the outcome of PI3K by dephosphorylating PIP3 to PIP2. PIP2 is converted to DAG and IP3 by phospholipase C (PLC), leading to Ca^2+^ release by IP3 and further PKC activation by DAG. PKC also activates the MAPK pathway. PKC, MEK1/2 and p38 phosphorylate the cytosolic components of NADPH oxidase (p47^phox^, p67^phox^ and p40^phox^) which translocate to membranes assembling the NADPH oxidase complex in the plasma membrane and specific granules, enabling ROS formation. Hydrogen peroxide (H_2_O_2_) activates and dissociates a complex formed by granule proteins, namely myeloperoxidase (MPO), neutrophil elastase (NE), azurocidin (AZU), cathepsin G (CG), lactoferrin (LTF), proteinase 3 (PR3) and lysozyme (LYZ). NE dissociation degrades F-actin allowing the translocation of NE to the nucleus to degrade histones. Moreover, PIP3 activates PKC ζ, which in turn, activates PAD4, causing the citrullination of histones. Akt, PAD4 and p38 activate NF-κB, releasing IL-1β coordinated with the gasdermin D cleavage by NE, allowing the release of NETs. See more details in the Appendix A.

## Data Availability

Not applicable.

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
