# Peer review of "Neutrophil Activated by the Famous and Potent PMA (Phorbol Myristate Acetate)"

_cells, 2022, doi:10.3390/cells11182889_

Round 1
Reviewer 1 Report
Congrats for this nice review. It is an interesting article. I agree with you that this is a growing field of interest as this is documented by the growing number of relevant studies in the literature. An article whose findings are important to those with closely related research interests. Although I recognize that the paper is well written and correct from a methodological point of view, I have a minor recommendation: do not use abbreviations in title. I do not have any additional comments.
Author Response
We are thankful to the reviewers, who dedicated their time and effort to make valuable suggestions to our manuscript. We carefully analyzed and addressed every suggestion, as detailed below, and made the appropriate adjustments in the text, highlighted in the track changes and comments in the MS-Word file.
Reply:
As PMA is a very common abbreviation in the field, and used in the title of several publications, instead of just removing the abbreviation we added its description to the title.
Reviewer 2 Report
Manuscript by Damascena et al. is a thorough, comprehensive, highly mechanistic review on activation of neutrophils with phorbol myristate acetate (PMA). PMA is a well-known inducer of NETosis, a specific suicidal death type of neutrophils that allows them to immobilize and destruct of microorganisms. PMA is an activator of protein kinase C (PKC) that can trigger NET formation though a reactive oxygen species (ROS)-dependent pathway. The process induced by PMA is a well-characterized and serve as a model to study NETosis. Authors review in details mechanism of the activation process and highlitening unknown or controversial issues, giving a complete compendium of current knowledge in this subject. The manuscript is suitable for publication in Cells after considering the below comments and suggestions:
1. Figure 2 looks poor quality. At least in a copy I have been reading. Please redraw the figure with higher quality.
2. Please correct minor spelling mistakes.
3. What I miss in this review article is the PMA-induced NETosis model references to real-life clinics and diagnostics. I know that the aim of the review was to report the intracellular PMA-induced processes in neutrophils, but I think it would be reasonable to at least add a short paragraph in the summary about the importance of research using PMA to understand what goes on in the patient's body during neutrophil activation and how it helps including activation study using PMA. Without it, this artificial model seems to have no valid biological, medical context.
Author Response
We are thankful to the reviewers, who dedicated their time and effort to make valuable suggestions to our manuscript. We carefully analyzed and addressed every suggestion, as detailed below, and made the appropriate adjustments in the text, highlighted in the track changes and comments in the MS-Word file.
- Figure 2 looks poor quality. At least in a copy I have been reading. Please redraw the figure with higher quality.
Reply:
Although the original image was sent as a high quality file (300 dpi, 27.39 x 17.07 cm, TIFF), here we provide another version at even higher quality (600 dpi).
- Please correct minor spelling mistakes.
Reply:
The new version of the text was revised by a native English speaker.
- What I miss in this review article is the PMA-induced NETosis model references to real-life clinics and diagnostics. I know that the aim of the review was to report the intracellular PMA-induced processes in neutrophils, but I think it would be reasonable to at least add a short paragraph in the summary about the importance of research using PMA to understand what goes on in the patient's body during neutrophil activation and how it helps including activation study using PMA. Without it, this artificial model seems to have no valid biological, medical context.
Reply:
The suggested information was added to the topic discussing NETs (page 8, lines 357-361).
NETs are a mechanism used by neutrophils, in which they release DNA and proteins to promote the extracellular capture and death of microorganisms. However, they are considered double-edge swords for being associated with the cause and severity of some diseases, such as thrombosis, autoimmune diseases, cancers. Recently, it was also associated with the severe acute respiratory syndrome observed in COVID-19 [136,137]. A better understanding of the pathways related to NETs is needed so that more effective therapeutic approaches to these diseases can be developed [135].
Reviewer 3 Report
The authors review in detail the results, obtained in many years of research, by employing PMA as a neutrophil stimulus. The purpose is very good and ambitious, considering the complexity of the mechanisms triggered by PMA and the frequent contrasting results reported by different authors. This review in my opinion deserves to be published, provided that the following comments will be addressed.
General comments:
PMA is one of the active compounds of croton tiglium oil, which was widely used in biological research since many years. Croton oil is a very active substance, which was widely used to study the biology of inflammation and carcinogenesis. Tumor promotion activity was traced to phorbol esters present in croton oil. Pure phorbol 12-myristate 13-acetate, which is found in croton oil is now used widely in laboratory research to promote tumor development and to dissect the mechanisms of leukocyte activation. I think that the discovery of PMA and its early use should be at least briefly mentioned.
Neutrophil degranulation, which is, together with the ingestion process, a very important function of these cells is only considered superficially. Of note importance is given to the possible heterotypic fusion between primary and specific (secondary) granules (see comment below). In my opinion the process of degranulation, to the best of my knowledge, is stimulated by PMA, and gave in the past contrasting results. This should be mentioned and discussed.
Frequently in the text acronims are used without explanation, which is reported in subsequent paragraphs (e.g. page 2 line 50 PAD4). In my opinion the explanation must be given at the first citation, together with a “see below” or see the paragraph…… when necessary.
The term “PMA induced cells” is not adequate, I think that PMA-stimulated/activated cells is more appropriate
References are listed by numbers, however at page 6 line 293 two of them are reported by name (Gupta and Suh). Please correct
Some paragraphs are no more than a simple list of results obtained by different authors (e.g. lines 155-185). I recommend the authors to include at the end of each paragraph a summary, easily to undestand, to help the reader’s comprehension.
SPECIFIC COMMENTS
Page 1, Line 39
A structural analogue (or simply analogue) is a compound having a structure similar to that of another compound, but differing from it in respect to certain component/s. From a chemical point of view DAG is hardly considered an analogue of PMA, as it can be easily deduced by. considering the structures of these compounds. Please correct
Page 3 line 122
E-sel and P-sel are endothelium surface molecules and not neutrophils molecules (L-sel is a neutrophil molecule). This must be precised.
Page 3 lines 151-153:
Please explain better why….. In addition to the fact ……………….Ros productions (19, 21, 56) reinforces the idea that the regulation of cytoskeleton remodeling is established by these isoforms of PKC.
Page 5 Legend to Fig.2
Line 227:
PIP3 cannot phosphorylate by itself
Lines 229-231
MPO, azurocidin, lactoferrin and lysozyme are not granule proteinases. They are indeed granule components and are released (is better than dissociated) upon PMA challenge (in different amount with respect to other agonists). Please correct.
Page 5 lines 237-
Only PI3K belonging to class I has been discussed. Also class II and III should be mentioned.
Page 5 line 250
The class I PI3K….activity
Page 5 line 246
PIP3 and not PI3P
Page 6 Lines 267:
a (see below) should be added
Page 6 Lines 271-272
PMA induced calcium influx or efflux
Page 6 Lines 288-292
As already reported in the General Comments, the process of heterotypic fusion between primary and secondary granules, waits to be confirmed and only a brief report (2016), to the best of my knowlege, supported its existance. No ultrasctructural evidence has been reported about it. In my opinion it is not necessary to call it in cause to explain intracellular ROS production, as cell membranes are completely permeable to hydrogen peroxide. So, I suggest to modify adequately the paragraph, by suggesting that heterotypyc fusion may be carried out, but that, at the same time ROS, may independently of this process, reach the cell interior.
Page 6 Lines 306
The acronim NET should be reported
Page 7 Lines 310-311:
I think that this is a truncated sentence.
Page 7 Lines 323:
In this sentence the glutathionylation of actin is claimed to lead to polymeriazation, while, this process is involved in actin depolymerization according to the sentence cited on line 142. This discrepancy should be explained.
Page 7 Lines 366:
citrullination or IL1b..... should be better
Page 8 Lines 367:
this sentence should be made more clear
Page 8 Lines 375-394:
in my opinion lines 385-394 should be moved above after line374.
Page 9 Lines 428:
(168),… allowing the release of …..
Page 9 Lines 429:
IL1b is involved in the process of …..
Page 9 Lines 430:
please explain what a large pathogen is
Page 9 Lines 458:
PAD4. However……
Page 9 Lines 462-463:
leading to both pore formation and leading to NET formation…..
and is involved in the process of NET formation,……………….
Author Response
We are thankful to the reviewers, who dedicated their time and effort to make valuable suggestions to our manuscript. We carefully analyzed and addressed every suggestion, as detailed below, and made the appropriate adjustments in the text, highlighted in the track changes and comments in the MS-Word file.
Reviewer 3
The authors review in detail the results, obtained in many years of research, by employing PMA as a neutrophil stimulus. The purpose is very good and ambitious, considering the complexity of the mechanisms triggered by PMA and the frequent contrasting results reported by different authors. This review in my opinion deserves to be published, provided that the following comments will be addressed.
General comments:
PMA is one of the active compounds of croton tiglium oil, which was widely used in biological research since many years. Croton oil is a very active substance, which was widely used to study the biology of inflammation and carcinogenesis. Tumor promotion activity was traced to phorbol esters present in croton oil. Pure phorbol 12-myristate 13-acetate, which is found in croton oil is now used widely in laboratory research to promote tumor development and to dissect the mechanisms of leukocyte activation. I think that the discovery of PMA and its early use should be at least briefly mentioned.
Reply:
We added the requested information to the introduction, lines 34-39.
Neutrophil degranulation, which is, together with the ingestion process, a very important function of these cells is only considered superficially. Of note importance is given to the possible heterotypic fusion between primary and specific (secondary) granules (see comment below). In my opinion the process of degranulation, to the best of my knowledge, is stimulated by PMA, and gave in the past contrasting results. This should be mentioned and discussed.
Reply:
A new topic was added (pages 2-3, lines 124-155)
Frequently in the text acronims are used without explanation, which is reported in subsequent paragraphs (e.g. page 2 line 50 PAD4). In my opinion the explanation must be given at the first citation, together with a “see below” or see the paragraph…… when necessary.
Reply:
We searched the text for unexplained acronyms and inserted the proper explanations. The example of PAD4 was corrected at Page 2, line 56.
The term “PMA induced cells” is not adequate, I think that PMA-stimulated/activated cells is more appropriate
Reply:
The term was replaced in the entire manuscript, as requested, e.g. Page 3, line 111
References are listed by numbers, however at page 6 line 293 two of them are reported by name (Gupta and Suh). Please correct
Reply:
All references are numbered now. Even though some of them are also mentioned by name in the text, the respective number was included.
Some paragraphs are no more than a simple list of results obtained by different authors (e.g. lines 155-185). I recommend the authors to include at the end of each paragraph a summary, easily to undestand, to help the reader’s comprehension.
Reply:
A summary of the most relevant concepts was included to such paragraphs, e.g. page 5, lines 213-215.
SPECIFIC COMMENTS
Page 1, Line 39
A structural analogue (or simply analogue) is a compound having a structure similar to that of another compound, but differing from it in respect to certain component/s. From a chemical point of view DAG is hardly considered an analogue of PMA, as it can be easily deduced by. considering the structures of these compounds. Please correct
Reply:
The question was addressed by avoiding the term analogue. Now at page 1, lines 43-44:
PMA is widely used to mimic PKC regulation by the same mechanism as diacylglycerol (DAG), …
Page 3 line 122
E-sel and P-sel are endothelium surface molecules and not neutrophils molecules (L-sel is a neutrophil molecule). This must be precised.
Reply:
The text was adjusted (now at page 4, lines 159-162)
Moreover, it decreases E-selectin [53,56] and P-selectin [57] on the surface of endothelial cells, fa-voring neutrophil adhesion to the endothelium.
Page 3 lines 151-153:
Please explain better why….. In addition to the fact ……………….Ros productions (19, 21, 56) reinforces the idea that the regulation of cytoskeleton remodeling is established by these isoforms of PKC.
Reply:
The text was adjusted (now at page 4, lines 189-194), further explaining the question.
Niggli et al. (1999) showed that the inhibitor of actin polymerization (latrunculin A) reduced cy-toskeletal PKCβII by 43%, suggesting a physical binding between PKCβII and F-actin [20]. In ad-dition, the fact that PKCζ inhibited PMA-activated adhesion and not ROS prodution suggests its possible effector function to trigger pathways leading adhesion and not ROS [19,21,73], rein-forcing the idea that the regulation of cytoskeleton remodeling is established by these isoforms of PKC.
Page 5 Legend to Fig.2
Line 227:
PIP3 cannot phosphorylate by itself
Reply:
Although we agree with the reviewer that this statement would be a mistake, we did not find any text stating that PIP3 phosphorylates itself.
Lines 229-231
MPO, azurocidin, lactoferrin and lysozyme are not granule proteinases. They are indeed granule components and are released (is better than dissociated) upon PMA challenge (in different amount with respect to other agonists). Please correct.
Reply:
The text was revised, we agree that the term “proteases” was incorrect. Eve though, we kept the term dissociation, as there is evidence in the literature of the presence of a protein complex in the granular membrane (Metzler et al., 2014 - doi: 10.1016/j.celrep.2014.06.044). After revised, the text at page 6, lines274-275 now states: Hydrogen peroxide (H2O2) activates and dissociates a complex formed by granule proteins, namely myeloperoxidase (MPO),…
Page 5 lines 237-
Only PI3K belonging to class I has been discussed. Also class II and III should be mentioned.
Reply:
Class II and III PI3K were included in this topic at pages 6-7, lines 293-302:
PI3K participates in the intracellular ROS production by the NADPH oxidase complex [40,48,106,107]. Class III PI3K phosphorylates PtdIns to PtdIns3P (PI3P), described to regulate ly-sosomal trafficking and to control autophagy. Interestingly, PI3P contributes to the first phase of intracellular ROS production by binding to the subunit p40phox of the NADPH oxidase [108,109]. Class II PI3Ks also phosphorylate PtdIns to PtdIns3P, however their function has been little ex-plored [100].Class I and II PI3K have a intrisec relation to ROS production, include to later acti-vation of proteins.Chen et al. (2010), showed that inhibition of Akt and deletion of the Akt2 gene decreased p47phox phosphorylation and translocation to the plasma membrane[45]. Akt also phosphorylates mTORC, inhibits anti-apoptotic proteins and activates the NF-B complex [104].However, more studies is necessary to elucidation about Class II PI3K.
Page 5 line 250
The class I PI3K….activity
Reply:
Corrected - page 7, line 303:
Page 5 line 246
PIP3 and not PI3P
Reply:
The intended metabolite here was actually phosphatidylinositol 3-phosphate (PtdIns3P or PI3P), not phosphatidylinositol 3,4,5 trisphosphate (PtdIns(3,4,5)P3 or PIP3). The text was rephrased to avoid confusion between similar abbreviations. Page 6, lines 294-296:
Class III PI3K phosphorylates PtdIns to PtdIns3P (PI3P), described to regulate lysosomal trafficking and to control autophagy. Interestingly, PI3P contributes to the first phase of intracellular ROS production by binding to the subunit p40phox of the NADPH oxidase [108,109].
Page 6 Lines 271-272
PMA induced calcium influx or efflux
Reply:
Corrected - page 7, line 323:
Page 6 Lines 288-292
As already reported in the General Comments, the process of heterotypic fusion between primary and secondary granules, waits to be confirmed and only a brief report (2016), to the best of my knowlege, supported its existance. No ultrasctructural evidence has been reported about it. In my opinion it is not necessary to call it in cause to explain intracellular ROS production, as cell membranes are completely permeable to hydrogen peroxide. So, I suggest to modify adequately the paragraph, by suggesting that heterotypyc fusion may be carried out, but that, at the same time ROS, may independently of this process, reach the cell interior.
Reply:
The topic of a possible heterotypic granule fusion and the access of ROS across membranes was removed from here (just a brief mention left) and discussed in the new topic about degranulation. page 7, lines 339-341:
Page 6 Lines 306
The acronim NET should be reported
Reply:
Corrected - page 8, line 355:
Neutrophil Extracellular Trap (NET) formation
Page 7 Lines 310-311:
I think that this is a truncated sentence.
Reply:
Corrected - page 8, line 363:
Suicidal NETs are ROS-dependent and vital NETs can be triggered by ROS-dependent or ROS independent mechanisms [135,140].
Page 7 Lines 323:
In this sentence the glutathionylation of actin is claimed to lead to polymeriazation, while, this process is involved in actin depolymerization according to the sentence cited on line 142. This discrepancy should be explained.
Reply:
Corrected - page 8, line 376:
In addition, F-actin glutathionylation triggered by ROS also leads to actin degradation [148], promoting structural changes
Page 7 Lines 366:
citrullination or IL1b..... should be better
Reply:
Corrected - page 9, line 419:
based on histone H3 citrullination or IL-1β release
Page 8 Lines 367:
this sentence should be made more clear
Reply:
The explanation was further detailed - page 9, line 418-421:
we suggest that PAD4 may influence NET formation by at least two pathways, based on histone H3 citrullination or IL-1β release and gasdermin cleavage triggered by activity of serine proteases. This second pathway possibly explains the presence of NETs in the absence of citrullinated histones, as the same proteases might cleave the citrullinated histones.
Page 8 Lines 375-394:
in my opinion lines 385-394 should be moved above after line374.
Reply:
Corrected - page 9, lines 429-437
Page 9 Lines 428:
(168),… allowing the release of …..
Reply:
Corrected - page 10, line 481
Page 9 Lines 429:
IL1b is involved in the process of …..
Reply:
Corrected - page 10, line 482
Page 9 Lines 430:
please explain what a large pathogen is
Reply:
Explained in the text - page 10, lines 483-484
in response to large pathogens that cannot be phagocytosed [189], also entrapped in NETs [139]
Page 9 Lines 458:
PAD4. However……
Reply:
Corrected - page 10, line 510
Page 9 Lines 462-463:
leading to both pore formation and leading to NET formation…..
and is involved in the process of NET formation,……………….
Reply:
Corrected - page 10, line 513-516
Round 2
Reviewer 3 Report
The authors exaustively addressed the reviewer's comments. However before publication two points should be corrected. 1: lines 179-180 of the revised version. Selectins are involved in neutrophil adhesion and rolling, and participate in the neutrophil migration process. Even if they are expected to be downregulated during this process, stating that their decrease can favour neutrophil adhesion is misleading. The sentence must be made more clear or corrected. 2. Legend to Fig. 2 In my comment I precised that PIP3 is not a kinase, as it seems to be stated in the legend. Please correct.
Author Response
We thank the reviewer for making further suggestions and improve the manuscript. Both suggestions were accepted and the text was modified accordingly.
Regarding the specific points:
1- Lines 179-180 of the revised version. Selectins are involved in neutrophil adhesion and rolling, and participate in the neutrophil migration process. Even if they are expected to be downregulated during this process, stating that their decrease can favour neutrophil adhesion is misleading. This sentence must be made more clear or corrected.
Reply:
We agree that the explanation was not clear and changed it accordingly. The new text is as follows:
Moreover, PMA triggers an increase of E-selectin on endothelial cells [52], while decreasing P-selectin glycoprotein ligand-1 (PSGL-1) on the surface of neutrophils, reducing their adhesion to P-selectin [53] on the surface of endothelial cells. These mechanisms of regulation are involved in the control of neutrophils rolling velocity on the endothelium, favoring their diapedesis.
2- Legend to Fig.2 In my comment I precised that PIP3 is not a kinase, as it seems to be stated. Please correct.
Reply:
We agree that indeed PIP3 is not an enzyme and, although involved in the process, it is not responsible for the phosphorylation. As the participation of PIP3 is better explained in the main text, we opted for just removing its mention here.